# Gut Microbiota Modulation of Moderate Undernutrition in Infants through Gummy *Lactobacillus plantarum* Dad-13 Consumption: A Randomized Double-Blind Controlled Trial

**DOI:** 10.3390/nu14051049

**Published:** 2022-03-01

**Authors:** Rafli Zulfa Kamil, Agnes Murdiati, Mohammad Juffrie, Endang Sutriswati Rahayu

**Affiliations:** 1Department of Food and Agricultural Product Technology, Faculty of Agricultural Technology, Universitas Gadjah Mada, Jl. Flora No 1 Bulaksumur, Yogyakarta 55281, Indonesia; raflizulfakamil@lecturer.undip.ac.id (R.Z.K.); amurdiati@ugm.ac.id (A.M.); 2Centre for Food and Nutrition Studies, Universitas Gadjah Mada, Jl. Teknika Utara Barek, Yogyakarta 55281, Indonesia; 3Centre of Excellence for Probiotics, Universitas Gadjah Mada, Jl. Teknika Utara Barek, Yogyakarta 55281, Indonesia; 4Department of Food Technology, Faculty of Animal and Agricultural Sciences, Universitas Diponegoro, Jl. Prof. Soedarto, Tembalang, Semarang 50275, Indonesia; 5Faculty of Medicine, Public Health and Nursing, Universitas Gadjah Mada, Jl. Farmako, Senolowo, Sekip Utara, Yogyakarta 55281, Indonesia; mjuffrie@ugm.ac.id

**Keywords:** gummy probiotic, *L. plantarum* Dad-13, moderate undernutrition, gut microbiota modulation, Short-Chain Fatty Acid

## Abstract

Undernutrition is associated with gut microbiota unbalance, and probiotics are believed to restore it and improve gut integrity. A randomized double-blind controlled trial was conducted to evaluate the efficacy of gummy *L. plantarum* Dad-13 (10^8−9^ CFU/3 g) to prevent the progression of severe undernutrition. Two groups of moderate undernutrition infants were involved in this study, namely the placebo (*n* = 15) and probiotics (*n* = 15) groups, and were required to consume the product for 50 days. 16S rRNA sequencing and qPCR were used for gut microbiota analysis, and gas chromatography was used to analyze Short-Chain Fatty Acid (SCFA). The daily food intake of both groups was recorded using food records. Our results revealed that the probiotic group had better improvements regarding the anthropometry and nutritional status. In addition, *L. plantarum* Dad-13 modulated the butyric acid-producing bacteria to increase and inhibit the growth of Enterobacteriaceae. This gut modulation was associated with the increment in SCFA, especially total SCFA, propionic, and butyric acid. The number of *L. plantarum* was increased after the probiotic intervention. However, *L. plantarum* Dad-13 was not able to change the alpha and beta diversity. Therefore, *L. plantarum* Dad-13 has been proven to promote the growth of beneficial bacteria.

## 1. Introduction

As a developing country, Indonesia is facing a double burden of malnutrition. According to the Indonesia Ministry of Health data, the number of infants with undernutrition exceeds that of infants with overnutrition [1]. Therefore, the management of undernutrition is prioritized rather than overnutrition. Undernutrition is classified as stunting (low height-for-age), wasting (low weight-for-age), and underweight (low weight-for-height) [2]. According to the Z-score’s cut-off value, the severity of undernutrition is classified as moderate (between −2 and −3 SD) and severe (<−3SD) malnutrition [2].

Furthermore, 17.7% of children in Indonesia suffer from wasting, of which 13.8% and 3.9% are moderate and severe, respectively [1]. Children with undernutrition may experience delayed growth and deficiency in energy, proteins, and micronutrients. They also have a higher risk of cognitive and motor developmental impairments. In addition, improving the health of children under five years old is the golden key to creating remarkable human resources. Therefore, the consequences of undernutrition incidence are undesirable.

Several published studies have indicated that perturbation of gut microbiota composition occurs in children with undernutrition [3,4,5,6], leading to the malabsorption of nutrients known as Environmental Enteric Dysfunction (EED) [7,8]. Our recent study showed that gut microbiota perturbation with Proteobacteria predominance occurred before the children were classified with severe undernutrition [9]. Children with moderate undernutrition also exhibit low stool Short-Chain Fatty Acid (SCFA) concentrations compared to normal children, in which SCFA is a vital regulator to maintain our gut’s health [9]. Therefore, the strategy to intervene in children with moderate malnutrition is one way to prevent the progression to severe malnutrition. According to Velly et al. [10], an intervention with antibiotics, prebiotics, and/or probiotics can restore gut microbiota perturbation toward normobiosis. However, antibiotics are less concerned due to their broad effect on the gut microbiota [11]. 

A probiotic is a living microorganism that, when adequately consumed, promotes the host’s health [12]. *L. plantarum* Dad-13 is an indigenous probiotic strain isolated from spontaneously fermented buffalo milk called “dadih”. This strain is known for its probiotic properties, such as resistance in gastrointestinal and antibacterial activity [13]. In addition, it has been assayed for its safety, in which no bacterial translocation occurred in the organs of the rat model [14]. The ability of probiotics to modulate the immune system and inhibit the growth of pathogenic bacteria may beneficially improve the anthropometry and nutritional status of children with undernutrition [15,16,17]. However, the lack of a gut microbiota analysis in the previous studies was a limitation. Therefore, to fill this knowledge gap, this study aimed to evaluate the gut microbiota modulation, anthropometry, and nutritional status improvement of infants with moderate undernutrition after an intervention with gummy *L. plantarum* Dad-13. In this study, gummy was used as a matrix for probiotic cells, as it can deliver the probiotic cells to pass gastrointestinal simulations [18].

## 2. Materials and Methods

### 2.1. Ethical Approval

The research protocol was approved by the Medical and Health Research Ethics Committee, Faculty of Medicine, Public Health, and Nursing, Universitas Gadjah Mada (approval date: 6 November 2019; reference number: KE/FK/1303/EC/2019) and was registered in https://www.thaiclinicaltrials.org/ (TCTR20220209009) (accessed on 9 February 2022) and https://ina-registry.org/ (INA-DC4CNNS) (accessed on 30 December 2021).

### 2.2. Sample Size Calculation

The sample size calculation followed the hypothesis tests for two population means (two-sided test) (Equation (1)) according to Lwanga and Lemeshow [19]:(1)$$n=\frac{2\delta^{2}{\left(Z_{1-\frac{\alpha }{2}}+Z_{1-\beta }\right)}^{2}}{{\left(µ1-µ2\right)}^{2}}$$
where n is the sample size, *δ* is the standard deviation of the population (assumed to be 0.95), *β* is the statistical power (assumed to be 10%), µ1 is the mean of the intervention group’s body weight increment (1.28 ± 0.94 kg; Surono et al. [17]), and µ2 is the mean of the control group’s body weight increment (0.99 ± 0.99 kg; Surono et al. [17]). The obtained sample size was then multiplied by the correction factor (lost-to-follow, assumed to be 20%). Therefore, 13 subjects for each group would be needed. 

### 2.3. Research Subjects and Randomization

The research was conducted in Tirtoadi Village, Sleman, Yogyakarta. A list of undernourished infants was obtained from public healthcare (Puskesmas Mlati II) and a home visit survey. They were prescreened according to their location, age, and the presence of a congenital disease. The infants’ parents or guardians who passed the prescreening were socialized based on research backgrounds. The parents or guardians who agreed to participate in the study signed the informed consent and assent forms for further screening. The inclusion criteria were having a Z-score cut-off between −2 and −3 standard deviation and not consuming probiotics, prebiotics, or antibiotics a month before the study. The infants who passed the screening were divided randomly into two groups: placebo and probiotics. Randomization was performed using Ms. Excel (2016) formulas = RAND() by an independent technician from the Centre for Food and Nutrition Studies. The research products were secretly coded for the placebo and probiotic groups. Both researchers and subjects did not know the product until the technician revealed it at the end of the study.

### 2.4. Research Product

Skim milk powder containing *L. plantarum* Dad-13 was used for gummy production and was produced by the Center for Food and Nutrition Studies, Universitas Gadjah Mada, according to Kamil et al. [20]. Gummy *L. plantarum* Dad-13 was produced according to the previous research by Kamil et al. [18]. The main ingredients consist of bovine gelatin, sucrose, glucose, water, and skim milk containing *L. plantarum* Dad-13. The gummy *L. plantarum* Dad-13 has viable cells 8.96 × 10^8^–1.16 × 10^9^ CFU/3g. The placebo product was also produced using the same formula. Instead of using skim milk containing *L. plantarum* Dad-13, skim milk powder (Lactona) was used. The gummy probiotic and placebo gross energy were 286.66 ± 0.88 [18] and 277.56 ± 1.12 kcal/100 g, respectively.

### 2.5. Research Design

This study was conducted from 21 January to 23 March 2020. A randomized double-blind controlled trial research design was used in this study for 90 days and was conducted with a per-protocol analysis approach. However, due to the constraints imposed by the COVID-19 pandemic, the intervention period was reduced to 50 days. Ten days before intervention, the subjects were prohibited from consuming prebiotics, probiotics, and laxatives to acclimate the gut condition. During the investigation, subjects were asked to consume gummy *L. plantarum* Dad-13 (3 g) once a day until the end of the study. The research design can be seen in Figure 1. 

### 2.6. Research Outcome 

During the intervention period, the daily food intake was recorded using a food record. The stool sample was collected before and after the study (±one day) for the gut microbiota, SCFA profile, and stool pH analysis. In addition, anthropometric measurement was conducted before and after the study to evaluate the nutritional status improvement. The primary outcomes of this research were subjects’ anthropometry and gut microbiota compositions. Meanwhile, the secondary outcomes were the dietary intake, SCFA profile, and stool pH.

#### 2.6.1. Anthropometric Measurement 

Bodyweight was measured using digital weight scales with the infant wearing a light cloth and no shoes (accuracy: 0.1 kg). Meanwhile, body height was measured using a 2-m-long microtoise without shoes (accuracy: 0.1 cm). The nutritional status of the subjects was calculated using the WHO Anthro 2005 program (https://www.who.int/toolkits/child-growth-standards/software) (accessed on 30 December 2021).

#### 2.6.2. Dietary Intake Analysis

Dietary intake was analyzed from food records. The type of food and its portion was input into the Nutrisurvey 2007 program, a nutritional calculation and survey program, an English version of German commercial software (EBISpro) (http://www.nutrisurvey.de/) (accessed on 30 December 2021). The nutritional intake was then calculated according to the Indonesian Food database downloaded on the same website. 

#### 2.6.3. Stool Sample Collection and DNA Extraction

During the intervention period, all subjects were asked to record their defecation time, frequency, and Bristol stool scale. Most subjects had normal Bristol stools (scale: 3 to 4). The stool samples were collected with the help of their parents or guardians. The subjects were asked to defecate on sterile trail paper while avoiding contamination from any water sources (urine or toilet water). A fresh stool sample was then scooped into two sterile container tubes. The first tube contained glass beads and 2 mL of RNA-later (Sigma-Aldrich; R0901; St. Louis, MO, USA) for the gut microbiota analysis. Meanwhile, the second tube was an empty tube for SCFA analysis. The collected stool samples were delivered to the laboratories within an icebox no more than 5 h from the defecation time and were immediately stored at −40°C until the analysis day. The Bead-beating method was used for DNA extraction according to Nakayama et al. [21], with modifications as previously described by Kamil et al. [9].

#### 2.6.4. DNA Quality Control and Purification for 16S rRNA Sequencing

All the PCR reactions were performed with the Phusion^®^ High-Fidelity PCR Master Mix (New England Biolabs, Boston, MA, USA). The obtained amplicon was then mixed with loading buffer (containing SYBR green) at the same volume, followed with 2% agarose electrophoresis for detection. Samples that had bright main strips ranging from 400 to 450 bp were chosen for further steps. The PCR products at equal density ratios were mixed and purified with the Qiagen Gel Extraction Kit (Qiagen, Hilden, Germany).

#### 2.6.5. 16S rRNA Sequencing

The sequencing was conducted only for selected subjects (randomly selected) from both groups (4 similar subjects for each group). However, one subject from the placebo group did not pass the DNA quality control (see Appendix A). 16S rRNA sequencing and data processing were performed by NovogeneAIT (Singapore), targeting the V3 to V4 variable regions (F (341F): CCTAYGGGRBGCASCAG; R (806R): GGACTACNNGGGTATCTAAT)). The libraries were generated with the NEBNext^®^ UltraTM DNA Library Prep Kit for Illumina (New England Biolabs, Boston, MA, USA) and were quantified via Qubit and qPCR. The 250 paired-end sequencing was performed on an Illumina HiSeq 2500 platform. 

#### 2.6.6. Sequencing Data Processing

The raw obtained sequence was merged using FLASH (V1.2.7) [22], followed with quality filtering according to the Quantitative Insights into Microbial Ecology (QIIME) (V1.7.0) quality control process [23]. The effective tag (removed chimera) was obtained with the UCHIME algorithm [24]. Uparse software (v7.0.1001) was used to analyze the effective tag [25], in which sequences with >97% similarity were assigned to the same OTUs. For the taxonomic annotation of each OTU representative, Mothur software was performed against the SSUrRNA database of the SILVA database (threshold: 0.8~1) [26]. Furthermore, the phylogenetic relationship of all OTUs representatives was obtained with MUSCLE (V 3.8.31) [27]. OTU abundance information was normalized using a standard sequence number corresponding to the sample with the least sequences. Alpha diversity indices (observed species, Chao1, Shannon, and Simpson) were used to analyze the complexity of the biodiversity. In addition, Beta diversity indices (weight and unweighted unifrac) were used to analyze species complexity. All these analyses were performed using QIIME (V1.7.0).

#### 2.6.7. qPCR Analysis

The qPCR analysis was conducted to calculate the absolute number (log10 bacterial cells/g stool) of interest bacteria *L. plantarum*, *Bifidobacterium*, and Enterobacteriaceae. Bio-Rad CFX96 (Bio-Rad, Berkeley, California, CA, USA) was used for qPCR analysis. The specific primer used for each bacterium can be seen in Table 1.

The sample was prepared by mixing 7-µL ddH2O (Otsu), 10-µL PCR mix (SMOBIO (ExcelTaqTM)), 1 µL of each forward and reverse primer, and 1-µL DNA template (DNA concentration was adjusted to 10 ng/µL). The standard curve was constructed by amplifying the single strain *L. plantarum* DNA ranging from 0.0001 to 50 ng/µL. 

#### 2.6.8. Stool pH and SCFA Analysis

The calibrated pH meter, pH Spear Eutech (Eutech Instruments, Paisley, United Kingdom), was directly dipped into the stool sample to measure the pH. Meanwhile, SCFA quantification was done as previously described by Kamil et al. [9]. In brief, 0.2 g of stool sample were diluted with 2-mL aquabidest, followed with sonication for 20 min. The supernatant was injected into GC (Shimadzu GC-2010 Plus) (Shimadzu, Kyoto, Japan) after centrifuging twice. 

### 2.7. Statistical Analysis

All the processed data were obtained from the infants who finished the study. The Wilcoxon rank-sum test was performed to analyze the difference between groups (placebo–probiotic). Meanwhile, the Wilcoxon paired test was performed to analyze the differences within groups (before–after intervention). A MetaStats analysis was conducted to identify gut microbiota composition differences via the nonparametric *t*-test, Fisher’s exact test, and false discovery rate. All those analyses were performed using R software (V2.15.3). The Linear Discriminant Analysis Affect Size (LEfSe) was conducted by LEfSe software to determine the overrepresentation of specific bacteria as biomarkers. Nonmetric Multidimensional Scaling (NMDS) based on Bray–Curtis dissimilarity was used to visualize the differences in microbial composition. In addition, the permutational multivariate analysis of variance (PERMANOVA) using Adonis function was used to analyze the significant differences. The results were considered significant at *p* < 0.001, *p* < 0.05, and *p* < 0.1.

## 3. Results

### 3.1. Demographic Data and Participant Flowchart

Forty infants passed the screening and were allocated for this study (20 infants for each group). However, only 15 subjects in each group finished the study. Five and four subjects in the placebo and probiotic groups resigned before the study began. One subject in the probiotic group did not collect the first stool sample. The subject participation flowchart is shown in Figure 2, and the subject characteristics are presented in Table 2.

### 3.2. Dietary Intake

Table 3 shows the nutrition intake between the two groups before and after the intervention. There were no significant changes in macronutrient and fiber intake in both groups. However, in the placebo group, an increment in the intake of vitamins E and C was observed, as well as a decrement in the vitamin K intake. Meanwhile, an increased intake of vitamins B1 and C was observed in the probiotic group. 

### 3.3. The Changes in Anthropometry and Nutritional Status

The changes in the anthropometry and nutritional status in both groups can be seen in Table 4. The weight and height in both groups increased significantly (*p* < 0.001). However, the increment of body weight in the probiotic group was higher than that in the placebo group, even though not significant when compared with the placebo group. An improvement in nutritional status was also observed in each group, but only the probiotic group had a significant improvement in all nutritional status categories (*p* < 0.05 and *p* < 0.1). Meanwhile, a significant improvement was only observed in the WAZ parameter for the placebo group (*p* < 0.05). Although nutritional status improvement was observed in the probiotic group, they were still categorized as having moderate malnutrition.

### 3.4. The Changes of Gut Microbiota Taxonomic between Groups

The 16S rRNA sequencing, targeting the V3 to V4 regions, produced a total high-quality read number of 1,512,897 (108,064.1 ± 25,956.43) and a total OTUs of 13,314 (951 ± 277.647). The taxonomy (top 10 relative abundance) of each group can be seen in Figure 3. Firmicutes, Bacteroidetes, Actinobacteria, and Proteobacteria were observed as the most dominant phylum in both groups. Furthermore, the major genus detected in both groups was *Prevotella_9*.

The figure shows that there is no significant effect of the placebo and probiotics at the phylum level. However, the intervention of probiotic *L. plantarum* Dad-13 tends to increase the number of the Firmicutes phylum (40.17–53.67%). In contrast, an expressive increment of the Bacteroidetes phylum was observed in the placebo group (30.51–47.36%). These relative abundance changes of the Firmicutes (*p*: 0.011) and Bacteroidetes (*p*: 0.005) phyla were significantly different if compared between groups according to the MetaStats analysis (see Appendix A).

The significant effect of the probiotic intervention was observed at the genus level, in which there was an increase of *Faecalibacterium* (9.71–15.34%; *p*: 0.029) and a decrease of *Agathobacter* (5.10–2.54%; *p*: 0.012) belonging to the Firmicutes phylum. The relative abundance change of *Faecalibacterium* in the probiotic group was significantly different from that of the placebo group (15.34% vs. 7.19%; *p*: 0.007) or was 2.13 times higher. In addition, *Prevotella_9* belonging to the Bacteroidetes phylum decreased twice (29.82% and 14.51%; *p*: 0.009) and was significantly lower than in the placebo (14.51% vs. 33.72%; *p*: 0.036). Other significant changes of a nondominant phylum and genus can be seen in Appendix A. The elevation of the genus-related Firmicutes phylum was also observed in the probiotics group, which were *Clostridium_sensu_stricto_1*, *Subdoligranulum*, *[Eubacterium]_hallii_group*, *[Eubacterium]_corprostanoligenes_group*, *Lachnospiraceae_NK4A136_group*, *Blautia*, and *Ruminococcus_2* (Figure 4). *Collinsella*, belonging to the Actinobacteria phylum, was also perceived for its increment after the probiotic intervention.

### 3.5. The Changes in Gut Microbiota Diversity and Composition

Alpha diversity reflects the gut microbiota richness, represented as observed species, and Chao1, Simpson, and Shannon indices, as shown in Figure 5A–D. Even though gut microbiota taxonomy changes were observed in the probiotic group, there were no significant changes in alpha diversity as calculated using the Wilcoxon and Tukey tests in all the indices (*p* < 0.05). Furthermore, beta diversity reflects the gut microbiota variation, which is calculated using weighted and unweighted unifrac. The weighted unifrac was determined based on the OTU abundance, and the unweighted unifrac was based on the phylogenetic relationship of the OTU. After 50 days of intervention, there was no significant difference between the placebo and probiotic groups regarding both beta diversity parameters (Figure 6A,B). It indicates that both groups’ gut microbiota community composition and relative abundance were not affected by the placebo or probiotic intervention, as statistically calculated using the Wilcoxon and Tukey tests (*p* < 0.05).

NMDS is a ranking method applicable to ecological studies. According to the given treatment (placebo and probiotic), it involves grouping the subject’s gut microbiota composition. The stress factor of NMDS was 0.065, which ensures the NMDS reliability result. As shown in Figure 7, the baseline in both groups was gathered closely on the negative axis of MDS1. In contrast, after the intervention of the placebo and probiotic, it was separated from the baseline on the positive axis of MDS1. In addition, according to the PERMANOVA (Adonis) analysis, an analysis of the grouping factor, and significance estimation based on a permutational test, a significant difference was detected between the placebo and probiotic groups after 50 days of intervention (*p*: 0.001) (Table 5). It indicates that probiotic intervention contributes to the changes in gut microbiota taxonomy in moderate undernourished infants.

### 3.6. Gut Microbiota Biomarker Identification

The LEfSe analysis was usually performed to determine the overrepresentation of specific bacteria in ecosystems as biomarkers [31]. This analysis emphasizes the statistical significance, biological relevance, and effect correlation. The results are shown as LDA scores and a cladogram (Figure 8A,B). The LDA score threshold was 4, and the length of each box represents the effect size. In addition, in the cladogram, the circle from inside to outside shows the phylum level of the genus. As indicated by the PERMANOVA analysis, a significant difference was only observed in both placebo and probiotic groups after 50 days of intervention, as shown from the LEfSe results.

As shown in Figure 8A,B, after 50 days of placebo intervention, only one genus was overrepresented, which was *Prevotella_2* (LDA score: 4.606; *p*: 0.034). In contrast, five genus overrepresentations were found in the probiotic group after 50 days of intervention, which were *Collinsella* (LDA score: 4.437; *p*: 0.034), belonging to the Actinobacteria phylum, and *Faecalibacterium* (LDA score: 4.587; *p*: 0.034), *Catenibacterium* (LDA score: 4.29; *p*: 0.032), *Subdoligranulum* (LDA score: 4.212; *p*: 0.034), and *Streptococcus* (LDA score: 4.261; *p*: 0.034), belonging to the Firmicutes phylum.

### 3.7. Specific Bacterial Quantification

*L. plantarum*, Bifidobacterium, and Enterobacteriaceae were selected as bacteria of interest, and their numbers were determined by qPCR analysis (Table 6). The quantification of *L. plantarum* aimed to evaluate its resistance in the gastrointestinal tract. Furthermore, according to our previous research, *Bifidobacterium* was found to be high in normal body weight infants [9]. In addition, Enterobacteriaceae represents potentially pathogenic bacteria. Table 6 shows the number of specific bacteria.

There were no significant changes in the number of *L. plantarum* and Enterobacteriaceae in the placebo group. However, the number of Bifidobacterium tended to decrease. In contrast, an expressive increment of *L. plantarum* and decrement of Enterobacteriaceae in the probiotic group were observed. However, there was no significant change in the number of *Bifidobacterium*.

### 3.8. SCFA Concentration and Stool pH

Table 7 shows the SCFA concentration and stool pH between the groups. A notable difference was observed in the probiotic group, mainly the elevation of total SCFA, propionic, and butyric acid concentrations after 50 days of intervention. In addition, a significant reduction of butyric acid was observed in the placebo group. However, the changes in SCFA concentration in both groups did not alter the stool pH significantly. 

## 4. Discussion

The restoration of the gut microbiota balance is the target for malnutrition treatment, since gut microbiota perturbation has been described in several studies [3,4,5,6]. Probiotic intervention is one of the alternatives. In this randomized double-blind controlled trial, *L. plantarum* Dad-13 was used and incorporated into gummy candy as a carrier. In addition, all the subjects were moderately stunted and wasted. Both groups had an insufficient intake of macronutrients (<70% RDA), especially energy, carbohydrate, and fat, whereas, according to the Indonesian RDA, the recommended intake of energy, carbohydrate, and fat for infants was 1350 (kcal), 215 (g), and 45 (g), respectively. In addition, the fiber intake in both groups was also less than 70% RDA, which was 19 (g). 

Even though the change of macronutrient intake in both groups was not significant, it tended to decrease and increase in the placebo and probiotic groups, respectively, which may affect the improvement of the anthropometry and nutritional status in the probiotic group. Research by Anukam et al. [32] suggested that probiotic intervention increases the appetite in rat models, supported by the study by Kazemi et al. [33] in which probiotic intervention also increased the energy intake in patients with depressive disorder. However, the relation between probiotic intervention and appetite or energy intake may differ depending on the subject’s physiology [33].

A review by Harahap and Suliburska [34] suggested that probiotic intervention may improve bone health, even though the paradoxical results and the mechanisms remain unclear. Besides that, the placebo group had a decrement intake of vitamin K. Vitamin K is essential for growth and mainly helps bone development (ossification), maintains bone density, and prevents the occurrence of osteoporosis. On the other hand, the probiotics group had an increment intake of vitamins B1 and C, which regulate growth and body metabolism [35,36]. Vitamin C also helps the absorption of iron that plays a role in bone development. Therefore, the improvement of the anthropometry and nutritional status in the probiotic group may be affected by the micronutrient intake.

The improvement of the anthropometry and nutritional status in the probiotic group aligns with a previous study by Surono et al. [17], although a different probiotic species and strain was used. The beneficial effect of probiotics to modulate the gut microbiota balance is by promoting the growth of beneficial bacteria that normally inhabits the intestine [37]. In this study, gut microbiota modulation tended to occur at the genus rather than at the phylum level. However, modulation did not commute the gut microbiota diversity and composition. This result aligns with the research by Gargari et al. [38], in which the alteration of alpha and beta diversity was not observed after a *Bifidobacterium bifidum* intervention in healthy adults. This result is probably due to the enormous size of the overall gut microbiota compared to the administered probiotics. In contrast, a study by Li et al. [39] indicated that a combination of *L. plantarum* LK006, *Bifidobacterium longum* LK014, and *B. bifidum* LK012p significantly reduced the alpha diversity indices (ACE, Chao1, Shannon, and Simpson) after 28 days of intervention in very low birth weight infants. The authors considered that the depletion of harmful bacteria, especially those belonging to the Proteobacteria phylum, affect the alpha diversity, whereas this was not observed in this study.

Cumulating data suggest that undernourished infants have a high relative abundance of Proteobacteria [3,4,5,6]. In addition, our recent research indicates that normal infants have a high relative abundance of Actinobacteria and Bacteroidetes and a low abundance of Proteobacteria [9]. The intervention of *L. plantarum* Dad-13 in moderately undernourished infants seemed not to have ameliorated the phylum level, as in normal infants. However, the administration of *L. plantarum* Dad-13 modulates the increase of the beneficial genus-related Firmicutes phylum. The elevation of the genus-related Firmicutes phylum was also observed in the study by Li et al. [39] and Castro-Mejía et al. [40] after administering probiotics. In addition, several genera belonging to the Firmicutes phylum have the ability to produce butyric acid, especially *Faecalibacterium*, *Roseburia*, *Butyrivibrio*, *Anaerostipes*, *Coprococcus*, *Oscillospira, Catenibacterium*, *Eubacterium*, *Ruminococcus*, *Clostridium*, and *Blautia* [41,42,43,44]. 

The gut microbiota modulation in the probiotics group was distinguishable from that in the placebo group, as indicated by the PERMANOVA result. It suggests that the probiotic treatment has a significant effect compared to the placebo on gut microbiota composition. In addition, the LEfSe analysis identified the overrepresented bacteria that distinguishes between the probiotic and placebo groups after the intervention. In the probiotics group, *Faecalibacterium*, *Catenibacterium*, *Subdoligranulum*, *Streptococcus,* and *Collinsella* were identified. *Faecalibaterium* is one of the genera that was identified as high in normal infants [10]. It is also known as the human source probiotic candidate, especially *Faecalibacterium prausnitzii* [45]. Several studies also mentioned its anti-inflammatory and immunomodulatory properties [41,46]. In addition, *Catenibacterium* is an obligate anaerobe bacterium that can produce not only butyric acid but also acetic, lactic, and iso-butyric acid [43]. This bacterium was also depleted in moderate acute malnutrition, especially *Catenibacterium mitsuokai* [5]. Our previous research also revealed that *Catenibacterium* was high in normal infants [9]. *Subdoligranulum* is also one of the butyric acid producers [44] that is less identified in Crohn’s disease cases [47].

On the other hand, *Streptococcus* has the ability to act as a probiotic, such as *Streptococcus thermophilus* [48]. It also exhibits immunomodulatory properties [49]. Even though *Collinsella* belongs to Actinobacteria, it also produces butyric acid [42] and is found high in normal infants [9]. In contrast, in the placebo group, *Prevotella_2* was identified. *Prevotella* is commonly found in the human intestine, especially in Indonesia, indicating the enterotype (P-type) [50,51]. In addition, the enrichment of *Prevotella stercorea* and *Prevotella copri* was associated with stunting incidence in a longitudinal birth cohort study in India [52].

To emphasize that gut microbiota modulation is affected by the probiotic intervention, we evaluated the number of specific bacteria. The increment in the number *L. plantarum* in the probiotic group indicated that *L. plantarum* Dad-13 could survive in the gastrointestinal tract, as described in other studies [13,53]. According to Velly et al. [10], undernourished infants lack beneficial bacteria such as *Bifidobacterium* and have an increase in potentially pathogenic bacteria (i.e., Enterobacteriaceae). In addition, probiotics intervention exhibits beneficial effects, promoting the number of beneficial bacteria and inhibiting pathogenic bacteria [41]. *L. plantarum* Dad-13 administration was able to inhibit Enterobacteriaceae growth. This result aligns with the study by Rahayu et al. [54]. Even though there was no significant effect on the number of *Bifidobacterium*, *L. plantarum* Dad-13 was able to promote butyric acid bacteria, as shown in the 16S rRNA sequencing results.

According to the investigation by Pekmez et al. [55], the SCFA concentration in infants with severe acute undernutrition was low, especially propionic and butyric acid. Thus, it is also observed in moderate undernutrition [9]. Additionally, during recovery, the SCFA concentration increases, along with the fecal bacterial number [55]. In parallel with the enrichment of the butyric acid producer, an elevation of total SCFA, propionic, and butyric acid in the probiotic group was observed after 50 days of intervention. *Faecalibacterium*, *Catenibacterium*, *Subdoligranulum*, and *Collinsella* have activity of acetyl-CoA acetyltransferase, acetyl/propionyl-CoA carboxylase, and butanol dehydrogenase, which contribute to butyric acid production [44]. In addition, *Subdoligranulum* exhibits glutaconyl-CoA decarboxylase activity, which is involved in butyric acid production from glutarate [44]. In contrast, in the placebo group, butyric acid was decreased. This indicates that there was no improvement in the gut environment of the placebo group, and also, the potentially pathogenic bacteria, Enterobacteriaceae, was highly found. Moreover, fiber intake as a substrate for producing SCFA was also decreased in the placebo group. According to Li et al. [56], organic acid production in the intestine affects the stool pH. In the probiotic group, the stool pH tends to be more acidic after 50 days of intervention. The acidification of the gut environment increases the bioavailability of Mg, Fe, and Ca [57].

It is suggested that SCFA, a product from the bacterial fermentation of nondigestible carbohydrates, is able to alter the energy metabolism and inhibit the pathogens and the adipogenesis process [55]. On the surface of intestinal epithelia are embedded SCFA-dependent receptors, which are free fatty acid receptor 3 (FFAR3/GPR41) and FFAR2 (GPR43). Notably, these receptors can also be found in white adipose tissue, skeletal muscle, and the liver [58]. It implies that SCFA might influence the substrate and energy metabolism in peripheral tissue. Furthermore, the activation of GPR41/43 maintains energy homeostasis through intestinal gluconeogenesis and ameliorates insulin sensitivity. According to Soty et al. [59], in malnutrition or low enteral intake, intestinal gluconeogenesis occurs approximately 20% higher than in the normal condition, which is only 5–7%. Therefore, the increment of SCFA, mainly propionic and butyric acid, acts as a substrate for gluconeogenesis. 

Moreover, butyric acid activates peroxisome proliferator-activated receptor gamma (PPRɣ) to maintain the hypoxia state of the intestine. The activation of PPRɣ manages colonocyte metabolism with regard to mitochondrial β-oxidation of fatty acids [60]. This mechanism suppresses the growth of facultative anaerobe pathogenic bacteria. It is also known that SCFA is involved in adipogenesis [61]. Butyric acid improved the activity of SREBP-1c (Streol Regulatory Element-Binding Protein 1c), which is a key regulator of adipogenesis. It also activates receptors in the stage of differentiation of adipogenesis, which are PPARγ, C/EBPα, and C/EBPβ. These explain the possible mechanisms of body weight increment after the probiotic intervention. In addition, *L. plantarum* Dad-13 has been proven to have the ability to produce folic acid, which is an essential micronutrient for growth [62].

Even though the intervention of gummy *L. plantarum* Dad-13 shows positive results, mainly in modulating butyric acid-producing bacteria, the low sample size and short intervention time were limitations of this study. The confounding factors such as physical activity, food availability, and supplement consumption may also affect the results. The use of the per-protocol analysis approach also led to bias, in which the analysis was only conducted for subjects who finished the research. In addition, in this study, the mechanism by which SCFA played the suggested role remains unclear. Hence, future validation studies are needed.

## 5. Conclusions

In conclusion, a 50-day intervention with gummy *L. plantarum* Dad-13 modulated the gut microbiota composition. It helped to improve the anthropometry and nutritional status of moderately undernourished infants. Gut microbiota modulation occurs at the genus level, and it mainly promotes the growth of butyric acid producer bacteria. Thus, it aligns with the increment of total SCFA, propionic, and butyric acid. The increment of the SCFA profile is suggested to be beneficial for energy balance, pathogen inhibition, and adipogenesis. Therefore, *L. plantarum* Dad-13 has the potential to prevent the progression of severe undernutrition in infants. 

## Figures and Tables

**Figure 1 nutrients-14-01049-f001:**
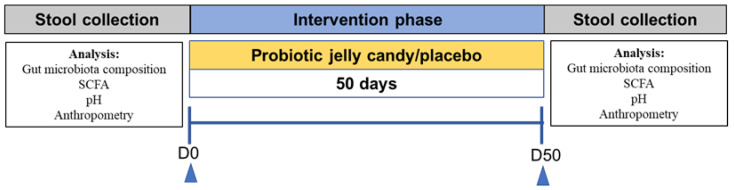
Research design. SCFA: Short-Chain Fatty Acid.

**Figure 2 nutrients-14-01049-f002:**
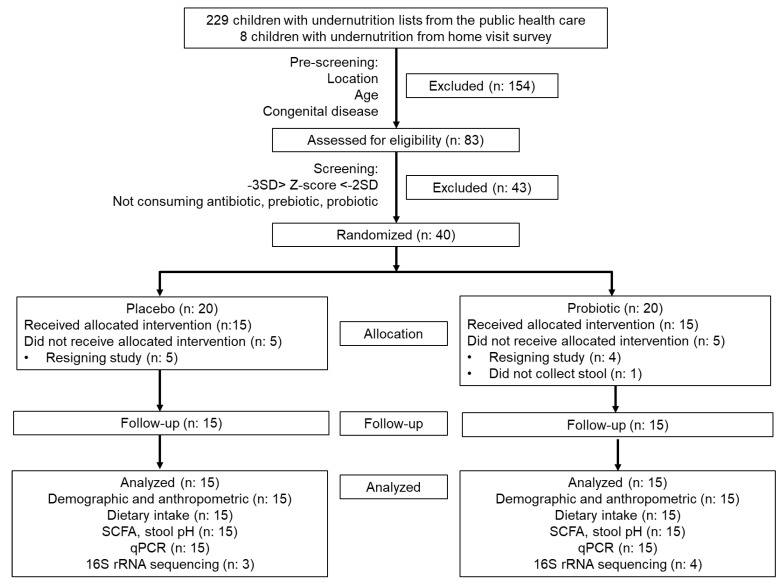
CONSORT diagram of subject participation during the study.

**Figure 3 nutrients-14-01049-f003:**
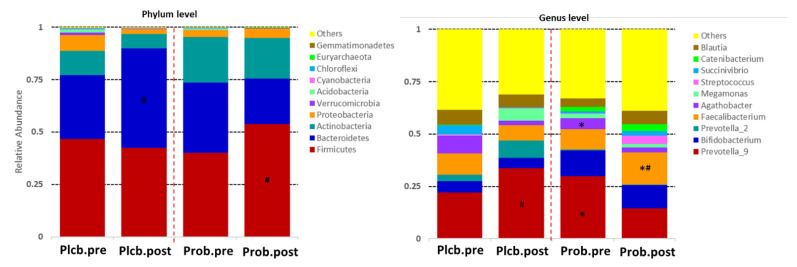
Top 10 relative abundance of gut microbiota composition between groups. ǂ PlcbPre–ProbPre; # PlcbPost–ProbPost; * ProbPre–ProbPost.

**Figure 4 nutrients-14-01049-f004:**
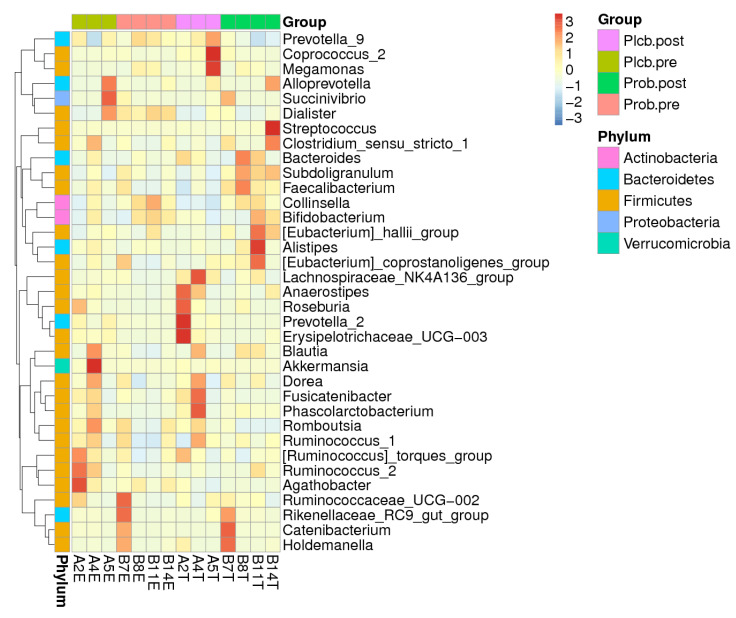
Heatmap of the top 35 relative abundance at the genus level of each subject.

**Figure 5 nutrients-14-01049-f005:**
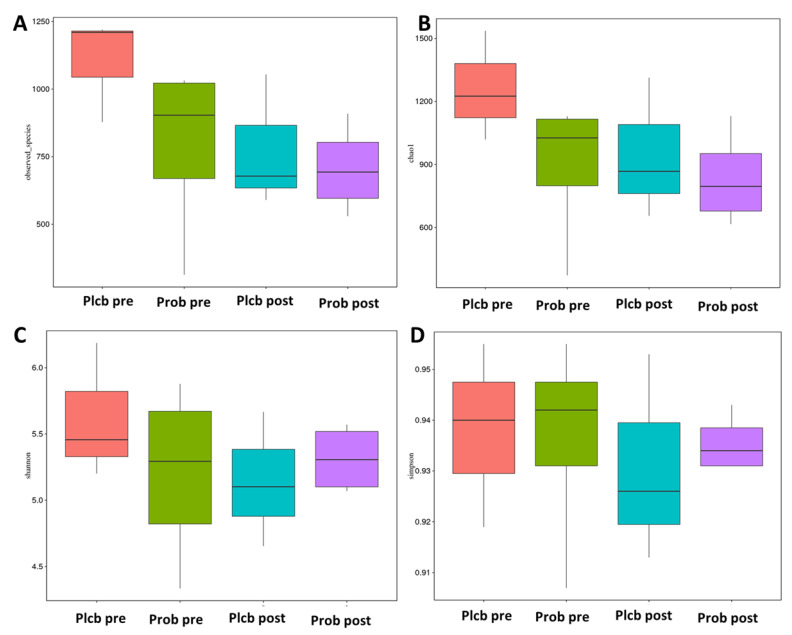
Boxplot alpha diversity index. (**A**) Observed species, (**B**) Chao1, (**C**) Shannon, and (**D**) Simpson.

**Figure 6 nutrients-14-01049-f006:**
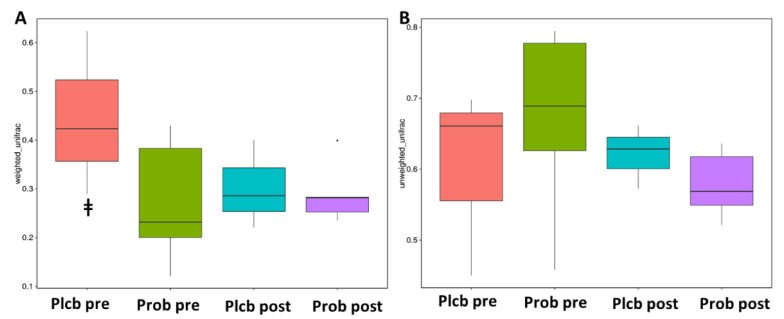
Box plot beta diversity index. (**A**) Weighted unifrac (**B**). Unweighted unifrac. ^ǂ^ PlcbPre–ProbPre.

**Figure 7 nutrients-14-01049-f007:**
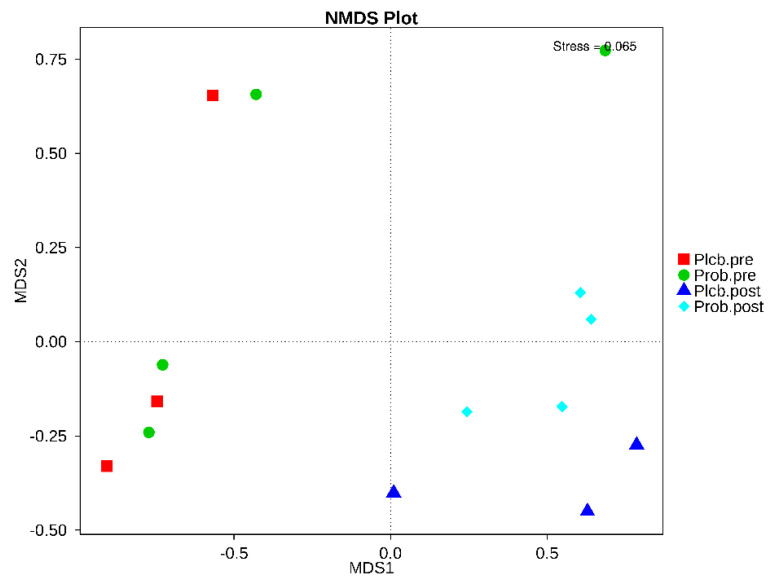
NMDS based on Bray−Curtis dissimilarity between the placebo and probiotic groups.

**Figure 8 nutrients-14-01049-f008:**
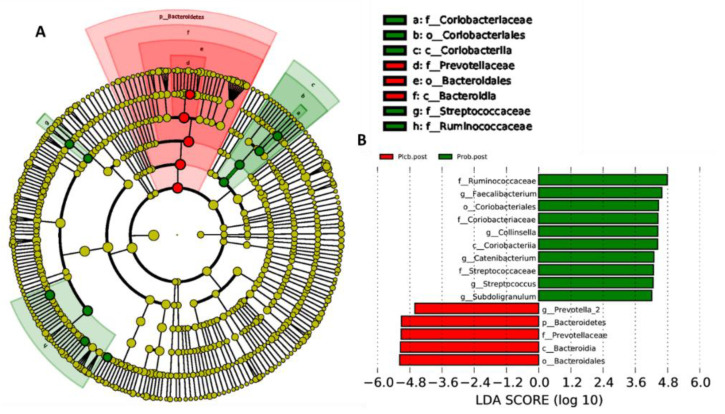
LEfSe analysis identified gut microbiota biomarkers between the placebo and probiotic groups after 50 days of intervention. (**A**) Cladogram and (**B**) LDA scores.

**Table 1 nutrients-14-01049-t001:** The specific primers used in this study.

Primer	5′–3′	Annealing (°C)	Ref
*Bifidobacterium*	g-Bifid-F CTCCTGGAAACGGGTGG g-Bifid-R GGTGTTCTTCCCGATATCTACA	58.8	[28]
*L. plantarum*	sg-Lpla-F CTCTGGTATTGATTGGTGCTTGCAT sg-Lpla-R GTTCGCCACTCACTCAAATGTAAA	60	[29]
Enterobacteriaceae	En-lsu-3F TGCCGTAACTTCGGGAGAAGGCA En-lsu-3’R TCAAGGACCAGTGTTCAGTGTC	60	[30]

**Table 2 nutrients-14-01049-t002:** Characteristics of the study subjects.

	Placebo (*n*: 15)	Probiotic (*n*: 15)	*p*
Male	10 (66.67%)	9 (60.00%)	
Female	5 (33.33%)	6 (40.00%)	
Age (months)	37.80 ± 11.78	37.93 ± 12.98	0.977
Weight (kg)	11.20 ± 1.96	10.84 ± 1.43	0.563
Height (cm)	88.88 ± 8.00	87.06 ± 6.84	0.509
WHZ	−1.40 ± 0.61	−1.19 ± 0.87	0.436
WAZ	−2.22 ± 0.74	−2.28 ± 0.94	0.838
HAZ	−2.21 ± 0.79	−2.55 ± 1.03	0.512

Data are presented as the mean ± SD. Wilcoxon rank-sum test (*p* < 0.05). WHZ: Weight for Height Z-score; WAZ: Weight for Age Z-score; HAZ: Height for Age Z-score.

**Table 3 nutrients-14-01049-t003:** Nutrition intake in the placebo and probiotic groups before and after the intervention.

	Unit	Placebo		*p*	Probiotic		*p*
		Before	After		Before	After	
Energy	kcal	677.13 ± 189.46	653.84 ± 185.74	0.733	681.99 ± 262.33	747.42 ± 263.18	0.427
Protein	g	27.58 ± 6.82	26.63 ± 7.79	0.570	27.39 ± 8.21	29.56 ± 10.57	0.670
Fat	g	26.13 ± 8.07	25.21 ± 8.38	0.638	26.74 ± 11.97	29.37 ± 12.26	0.320
Carbohydrate	g	83.15 ± 28.38	80.05 ± 25.79	0.776	82.93 ± 34.05	91.49 ± 33.36	0.363
Fiber	g	3.56 ± 1.73	3.17 ± 1.53	0.197	3.09 ± 1.37	3.13 ± 1.33	0.861
Vit. A	µg	386.71 ± 207.88	376.87 ± 140.36	0.955	584.82 ± 401.09	442.17 ± 240.49	0.532
Vit. E	mg	2.75 ± 1.40	3.33 ± 1.19	0.094	2.83 ± 1.33	3.43 ± 1.84	0.207
Vit. D	µg	2.83 ± 2.26	3.82 ± 2.09	0.152	3.17 ± 2.16	4.15 ± 3.06	0.147
Vit. B1	mg	0.28 ± 0.10	0.28 ± 0.10	0.971	0.25 ± 0.10	0.31 ± 0.14	0.058
Vit. B2	mg	0.50 ± 0.20	0.53 ± 0.18	0.558	0.57 ± 0.23	0.58 ± 0.25	0.969
Vit. B6	mg	0.43 ± 0.14	0.40 ± 0.12	0.371	0.41 ± 0.16	0.43 ± 0.14	0.587
Vit. K	µg	5.47 ± 3.76	3.22 ± 3.14	0.050	3.53 ± 1.48	2.71 ± 2.19	0.686
Folic acid	µg	75.93 ± 30.71	64.83 ± 20.56	0.211	89.82 ± 36.89	79.85 ± 36.92	0.649
Vit. C	mg	19.18 ± 16.25	33.08 ± 19.75	0.009	24.91 ± 19.07	35.46 ± 22.79	0.078
Na	mg	245.35 ± 124.63	298.76 ± 165.15	0.363	256.49 ± 110.42	323.35 ± 147.23	0.281
K	mg	681.55 ± 341.62	754.07 ± 274.48	0.363	747.73 ± 360.18	859.49 ± 411.01	0.460
Ca	mg	291.37 ± 251.24	379.77 ± 212.47	0.140	340.08 ± 246.67	443.14 ± 315.65	0.281
Mg	mg	96.25 ± 32.86	92.79 ± 30.55	0.460	93.69 ± 36.68	99.03 ± 37.24	0.460
P	mg	432.19 ± 180.76	471.27 ± 169.25	0.427	437.97 ± 188.58	515.21 ± 254.79	0.307
Fe	mg	5.29 ± 3.00	5.61 ± 2.54	0.670	6.52 ± 3.65	6.24 ± 3.49	0.615
Zn	mg	3.47 ± 1.08	3.35 ± 1.07	0.801	3.39 ± 1.17	3.77 ± 1.49	0.460

Data are presented as the mean ± SD. Wilcoxon paired test (*p* < 0.05 and *p* < 0.1).

**Table 4 nutrients-14-01049-t004:** Anthropometry and nutritional status of the placebo and probiotic groups before and after the intervention.

Parameter	Group	Before	After	*p*	Increment	*p*
Weight (kg)	Placebo	11.20 ± 1.96	11.59 ± 1.96	0.000	0.39 ± 0.30	0.109
	Probiotic	10.84 ± 1.43	11.43 ± 1.38	0.000	0.59 ± 0.36	
Height (cm)	Placebo	88.88 ± 8.00	90.17 ± 8.25	0.000	1.29 ± 0.68	0.980
	Probiotic	87.06 ± 6.84	88.35 ± 6.67	0.000	1.29 ± 0.75	
WHZ	Placebo	−1.40 ± 0.61	−1.30 ± 0.74	0.140	0.11 ± 0.39	0.187
	Probiotic	−1.19 ± 0.87	−0.90 ± 0.76	0.022	0.30 ± 0.48	
WAZ	Placebo	−2.22 ± 0.74	−2.04 ± 0.78	0.012	0.18 ± 0.24	0.187
	Probiotic	−2.28 ± 0.94	−2.01 ± 0.76	0.080	0.27 ± 0.30	
HAZ	Placebo	−2.21 ± 0.79	−2.04 ± 0.74	0.256	0.17 ± 0.27	0.806
	Probiotic	−2.55 ± 1.03	−2.35 ± 1.03	0.015	0.20 ± 0.26	

Data are presented as the mean ± SD. Wilcoxon rank-sum test, Wilcoxon paired test (*p* < 0.05 and *p* < 0.1). WHZ: Weight for Height Z-score; WAZ: Weight for Age Z-score; HAZ: Height for Age Z-score.

**Table 5 nutrients-14-01049-t005:** PERMANOVA analysis based on Bray−Curtis dissimilarity between the placebo and probiotic groups.

vs. Group	R^2^	*p*
PlcbPre–ProbPre	0.1519	0.646
PlcbPost–ProbPost	0.33456	0.001
PlcbPre–PlcbPost	0.21162	0.500
ProbPre–ProbPost	0.19063	0.265

R^2^: Grouping factor based on differences of the samples calculated from the ratios of grouping variance and total variance.

**Table 6 nutrients-14-01049-t006:** The number of specific bacteria analyzed by qPCR.

	Group	Log 10 Bacterial Cells/g Feces		*p*
		Before	After	
*L. plantarum*	Placebo	4.89 ± 0.32	4.89 ± 0.54	0.887
	Probiotic	4.85 ± 0.30	5.53 ± 0.79	0.027
*Bifidobacterium*	Placebo	6.24 ± 1.54	6.07 ± 0.84	0.087
	Probiotic	6.24 ± 1.21	6.50 ± 0.93	0.776
Enterobacteriaceae	Placebo	6.55 ± 0.68	6.28 ± 0.56	0.221
	Probiotic	6.27 ± 0.67	5.80 ± 0.76	0.027

Data are presented as the mean ± SD. Wilcoxon paired test (*p* < 0.05 and *p* < 0.1).

**Table 7 nutrients-14-01049-t007:** The changes of the SCFA concentration and stool pH between the groups after the intervention.

SCFA (mmol/g Feces)				
	Group	Before	After	*p*
Total SCFA	Placebo	35.83 ± 17.22	29.28 ± 15.26	0.185
	Probiotic	23.55 ± 9.03	33.78 ± 14.16	0.024
Acetic acid	Placebo	21.77 ± 12.07	17.41 ± 9.79	0.194
	Probiotic	15.28 ± 7.61	19.40 ± 7.63	0.156
Propionic acid	Placebo	6.57 ± 3.75	6.92 ± 4.70	0.930
	Probiotic	4.43 ± 2.46	6.89 ± 3.95	0.053
Butyric acid	Placebo	5.04 ± 2.64	3.56 ± 2.32	0.023
	Probiotic	2.62 ± 1.59	4.67 ± 2.95	0.017
Stool pH				
	Group	Before	After	*p*
pH	Placebo	6.23 ± 0.29	6.29 ± 0.35	0.607
	Probiotic	6.28 ± 0.28	6.10 ± 0.46	0.185

Total SCFA was the sum of acetic, propionic, iso-butyric, butyric, iso-valeric, valeric, and iso-caproic acid. Data are presented as the mean ± SD. Wilcoxon paired test (*p* < 0.05 and *p* < 0.1).

## Data Availability

The data presented in this study are available on request from the corresponding author. The data are not publicly available due to privacy protection.

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
