# Peer review of "Gut Microbiota Modulation of Moderate Undernutrition in Infants through Gummy Lactobacillus plantarum Dad-13 Consumption: A Randomized Double-Blind Controlled Trial"

_nutrients, 2022, doi:10.3390/nu14051049_

Round 1
Reviewer 1 Report
The submitted manuscript “Gut Microbiota Modulation of Moderate Undernutrition in Infants through Gummy Lactobacillus plantarum Dad-13 Consumption: A Randomized Double-Blind Controlled Trial” is interesting and focus on an important problem to human health.
The Tittle of the manuscript seems appropriate and reflects the content of the study.
The methodology is suitable, even though with some limitations.
Lines 148-149: “The stool samples were then stored at −40°C immediately within five hours until the analysis day.” The samples were stored immediately or within five hours? If there was a five hours interval until storage at -40ºC, were the samples refrigerated? This is important because the possibility of changes in the Microbiota, pH, etc.
Lines 159-166: “The sequencing was conducted only for selected subjects from both groups”. There was a selection criteria? Or were randomly selected?
16S rRNA sequencing only in 8 individuals? Can this be reliable for conclusions?
“However, one subject from the placebo group did not pass the DNA quality control”. Why this subject sample was not replaced by another?
The results are correctly exposed. However, some incongruities are present.
Lines 223: “There were no changes in macronutrient and fibre intake in both groups before and after the intervention.” In table 3, energy, protein, fat, carbohydrates and fiber increased in the probiotic group between the beginning and the end of intervention. The opposite occurs in the placebo group. These changes in nutrition intake can be responsible or influence the changes in weight, height and gut microbiota observed? It seems very hazardous assume that the observed changes were from Lactobacillus plantarum Dad-13 if the nutrition intake changed, even though below the nutritional requirements for infants of that age.
By the other hand, the weight and height improved in both groups, and the height growth was similar in both groups…
Figures 3, 4, 5 and 6 appear in the manuscript before the corresponding text. It was better to understand if they appear after the corresponding text.
In Figures 6 and 6 there is 1 Plcb pre, 1 Plcb post, 2 prob post and none prob pre. Is that correct?
Line 299: “In contrast, after the intervention of the placebo and probiotic, it was separated from the baseline on the positive axis of MDS1.” If after the intervention both groups are on the positive axis of MDS1, which is the probiotic effect?
The discussion is correct to the results.
The main problem is to assume that all the results are due to Lactobacillus plantarum Dad-13 consumption, but it seems there were changes in the nutrition intake, with differences between groups. Even though the Lactobacillus plantarum Dad-13 consumption could cause changes in gut Microbiota, the nutrition intake changes should be taken in consideration and discussed.
In general, the study should be improved/corrected in methodology and results sections.
Author Response
Dear Reviewer,
Dear Editorial team and Reviewer
Thank you for making quick progress on my manuscript. We appreciate your time and effort in providing valuable feedback on our manuscript. We have been able to incorporate changes to reflect most of the suggestions provided by the reviewers.
- Reviewer comment:
Lines 148-149: “The stool samples were then stored at −40°C immediately within five hours until the analysis day.” The samples were stored immediately or within five hours? If there was a five hours interval until storage at -40ºC, were the samples refrigerated? This is important because of the possibility of changes in the Microbiota, pH, etc.
Response:
Thank you for pointing this out. The sentence was revised:
The collected stool samples were delivered to the laboratories within the icebox no longer than 5 hours from the defecation time, and were immediately stored at −40°C until the analysis day (154-156)
- Reviewer comment:
Lines 159-166: “The sequencing was conducted only for selected subjects from both groups”. There was a selection criteria? Or were randomly selected?
Response:
The samples for sequencing analysis were randomly selected. However, for its reliable result, the samples were the same before and after treatment in both groups. (line 167-168)
- Reviewer comment:
16S rRNA sequencing only in 8 individuals? Can this be reliable for conclusions?
“However, one subject from the placebo group did not pass the DNA quality control”. Why this subject sample was not replaced by another?
Response:
We realize that the small number of samples for sequencing would be our research’s limitation, and we do consider this matter. However, we have a lack of financial support to conduct sequencing for all samples. Therefore, to anticipate nonreliable results, we tried to do sequencing for the same samples before and after treatment in both groups. We also analyzed the number of specific bacteria using qPCR (all samples) to support the sequencing results (Table 6), whether the probiotic intervention modulates gut microbiota composition, which represents the beneficial and potentially pathogenic bacteria according to the previous reference.
- Kamil, R.Z.; Murdiati, A.; Juffrie, M.; Nakayama, J.; Rahayu, E.S. Gut Microbiota and Short-Chain Fatty Acid Profile between Normal and Moderate Malnutrition Children in Yogyakarta, Indonesia. Microorganisms 2021, 9, 1–15, doi:10.3390/microorganisms9010127.
- Velly, H.; Britton, R.A.; Preidis, G.A. Mechanisms of Cross-Talk between the Diet, the Intestinal Microbiome, and the Undernourished Host. Gut Microbes 2017, 8, 98–112, doi:10.1080/19490976.2016.1267888.
- Effects of Consumption of Fermented Milk Containing Indigenous Probiotic Lactobacillus Plantarum Dad-13 on the Fecal Microbiota of Healthy Indonesian Volunteers. International Journal of Probiotics and Prebiotics 2016, 11, 91–98.
- Banin, M.M.; Utami, T.; Cahyanto, M.N.; Widada, J.; Rahayu, E.S. Effects of Consumption of Probiotic Powder Containing Lactobacillus Plantarum Dad-13 on Fecal Bacterial Population in School-Age Children in Indonesia. International Journal of Probiotics and Prebiotics 2019, 14, 1–8, doi:10.37290/ijpp2641-7197.14:1-8.
- Reviewer comment:
The results are correctly exposed. However, some incongruities are present.
Lines 223: “There were no changes in macronutrient and fibre intake in both groups before and after the intervention.” In table 3, energy, protein, fat, carbohydrates and fiber increased in the probiotic group between the beginning and the end of the intervention. The opposite occurs in the placebo group. These changes in nutrition intake can be responsible or influence the changes in weight, height and gut microbiota observed? It seems very hazardous assume that the observed changes were from Lactobacillus plantarum Dad-13 if the nutrition intake changed, even though below the nutritional requirements for infants of that age.
By the other hand, the weight and height improved in both groups, and the height growth was similar in both groups…
Response:
Thank you for the detailed review, we appreciate your suggestion and consider it as a factor that may affect the improvement of anthropometry and nutritional status. Therefore, we elaborate more in the discussion section with more references (Line 370-377).
- Reviewer comment:
Figures 3, 4, 5 and 6 appear in the manuscript before the corresponding text. It was better to understand if they appear after the corresponding text.
Response:
Revised as suggested
- Reviewer comment:
In Figures 6 and 6 there is 1 Plcb pre, 1 Plcb post, 2 prob post and none prob pre. Is that correct?
Response:
Thank you for noticing this error, the figures are revised.
- Reviewer comment:
Line 299: “In contrast, after the intervention of the placebo and probiotic, it was separated from the baseline on the positive axis of MDS1.” If after the intervention both groups are on the positive axis of MDS1, which is the probiotic effect?
Response:
The shifting axis in both groups before and after treatment may indicate the change in gut microbiota composition. In addition, we conduct the PERMANOVA analysis to ensure how significant the change was after the intervention. The PERMANOVA analysis, suggests that there was no significant change in both groups before and after the intervention, except between placebo and probiotic group after intervention. It suggests that there is a probiotic effect that distinguishes both groups after the intervention. (Line 307-312).
To easily access this note, the reviewer can download the documents attached.
Reviewer 2 Report
Major comments:
This study aimed to investigate the gut microbiota modulation, anthropometry, and nutritional status improvement of infants with moderate undernutrition after an intervention with a commercial product. The authors should address the following significant issues before further consideration.
1. Sample size estimation:
The β in the formula means statistical power, rather than power test (100 [1-β%]), according to the original article (https://apps.who.int/iris/handle/10665/40062). In addition, the assumed difference of body weight increment was not clearly described. This study's so-called minimal sample size (13) is probably wrong and too small to make adequate randomization.
2. Methods:
2-1: The preregistered information at https://ina-registry.org/index.php?act=registry_trial_detail&code_trial=09202131112337DC4CNNS was not enough for international investigators to repeat the results.
2-2: There are neither prespecified analytic plans, such as intention-to-treat analysis or per-protocol analysis.
Overall, the methods of this study were not academic enough, which precluded the accuracy of results and scientific soundness.
Author Response
REVIEWER 2
Dear Editorial team and Reviewer
Thank you for making quick progress on my manuscript. We appreciate your time and effort in providing valuable feedback on our manuscript. We have been able to incorporate changes to reflect most of the suggestions provided by the reviewers.
- Sample size estimation:
The β in the formula means statistical power, rather than power test (100 [1-β%]), according to the original article (https://apps.who.int/iris/handle/10665/40062). In addition, the assumed difference of body weight increment was not clearly described. This study's so-called minimal sample size (13) is probably wrong and too small to make adequate randomization.
Thank you for your detailed review, the mean of β value was revised, and the detailed average weight increment of bodyweight according to the reference was attached in the manuscript (line 83-93). In this research, we refer to the previous research by Surono et al [1], who conducted similar research although the used probiotic strain and food carrier were different. We aimed to evaluate whether the probiotic intervention can improve anthropometry in malnourished infants and also possible gut microbiota modulation which was not analyzed in previous research by Surono et al [1]. The obtained sample size according to the equation is 13 subjects for each group, however, in actuality, we recruit 20 subjects for each group, as can be seen in the subject’s participation diagram (line 228).
- Surono, I.S.; Koestomo, F.P.; Novitasari, N.; Zakaria, F.R.; Yulianasari; Koesnandar Novel Probiotic Enterococcus Faecium IS-27526 Supplementation Increased Total Salivary SIgA Level and Bodyweight of Pre-School Children: A Pilot Study. Anaerobe 2011, 17, 496–500, doi:10.1016/j.anaerobe.2011.06.003.
- Methods:
2-1: The preregistered information at https://ina-registry.org/index.php?act=registry_trial_detail&code_trial=09202131112337DC4CNNS was not enough for international investigators to repeat the results.
Thank you for pointing this out for the best quality of our research. The research protocol was already registered to the https://www.thaiclinicaltrials.org/ in accordance with the ICTRP’s (International Clinical Trials Registry Platform) list (https://trialsearch.who.int/). The registered ID was TCTR20220209009, and was attached to the manuscript. (line 80-81)
2-2: There are neither prespecified analytic plans, such as intention-to-treat analysis or per-protocol analysis.
This study was conducted with a per-protocol analysis approach, which compares the treatment in two groups placebo and probiotic, and only analyzes the subject who finished the study as described in Research design (line119-121) and statistical analysis (line 209). This would be also our research limitation, because the use of per-protocol analysis may lead to bias. Therefore, we also explain this as our research limitation (line 492-494).
Reviewer 3 Report
The authors propose a well-organized trial, the conclusions are well supported by the data obtained.
The idea of supplementation as a gummy is interesting and also the source of the administered bacteria is regional and immersed in the reality of the place of origin.
Limitations should be emphasized, in particular on the determination of bacterial lines which (although well performed) depend on how the feces are collected but also produced (for example the Bristol stool scale determines uniformity), as well as the methods must be well consolidated (for example 10.1007 / s13205-020-02351-w)
On the other hand, the results in terms of weight are evident so it can be concluded that the intervention is effective, perhaps it should be combined with nutritional intervention, first of all improving the general condition.
It would have been interesting to have a control group with subjects, not in undernutrition in order to verify the efficacy even in healthy subjects.
Author Response
REVIEWER 3
Dear Editorial team and Reviewer
Thank you for making quick progress on my manuscript. We appreciate your time and effort in providing valuable feedback on our manuscript. We have been able to incorporate changes to reflect most of the suggestions provided by the reviewers.
- Reviewer comment:
Limitations should be emphasized, in particular on the determination of bacterial lines which (although well performed) depend on how the feces are collected but also produced (for example the Bristol stool scale determines uniformity), as well as the methods must be well consolidated (for example 10.1007 / s13205-020-02351-w).
Response:
Thank you for your suggestion. During the intervention period, all the subjects were asked not only to fill out the food record but also to record the defection time and frequency, as well as the Bristol stool scale. Both groups were having a normal Bristol stool scale before and after intervention on average (scale 3-4; 70-80%). We add a little information about this in the stool sample collection and DNA extraction section (Line 147-148).
- Reviewer comment:
The results in terms of weight are evident so it can be concluded that the intervention is effective, perhaps it should be combined with nutritional intervention, first of all improving the general condition.
Response:
We really appreciate your idea and would be our consideration for the next. Thank you
- Reviewer comment:
It would have been interesting to have a control group with subjects, not undernutrition in order to verify the efficacy even in healthy subjects.
Response:
Our previous research was profiling gut microbiota in normal and moderate undernutrition Kamil et al [1]. In this research even though control (normal group) was not included, there were several bacteria that were found high in the normal group such as Catenibacterium and Collinsella to be increased after the probiotic intervention (line 428-429 & 433-434).
- Kamil, R.Z.; Murdiati, A.; Juffrie, M.; Nakayama, J.; Rahayu, E.S. Gut Microbiota and Short-Chain Fatty Acid Profile between Normal and Moderate Malnutrition Children in Yogyakarta, Indonesia. Microorganisms 2021, 9, 1–15, doi:10.3390/microorganisms9010127.
Reviewer 4 Report
Thank you very much to the Editor of NUTRIENTS for allowing me to review the paper entitled ‘Gut Microbiota Modulation of Moderate Undernutrition in Infants through Gummy Lactobacillus plantarum Dad-13 Consumption: A Randomized Double-Blind Controlled Trial’. Authors suggested that Lactobacillus plantarum Dad-13 has been proven to promote the growth of beneficial bacteria in infants.
Major comments:
This study is well-designed. The authors’ method is adequate, and the statistical analyses seem correct. Results were rather given clearly with sufficient tables and data analysis.
However, the statistics lack information about p-value as statistical significant (so how should be interpreted sentence: ”Wilcoxon paired test (p < 0.05 and p < 0.1).
Lack of information about the program used to evaluate the nutrition intake.
Significant results, connected with decreased butyric acid level after 50 days of using a placebo (5.04 ± 2.64 vs 3.56 ± 2.32; p=0.023) – have been without complete interpretation in DISCUSSION. (“In contrast, in the placebo group, butyric acid was decreased. This indicates that there was no improvement in the gut environment of the placebo.” Such a result suggests that maybe using milk as a placebo is not suitable for the development of infants.
Moreover, the paper lacks a paragraph entitled LIMITATIONS of study (at the end of DISCUSSION). In this paragraph, it should be included, for example – that the daily food intake of both groups was recorded using food records (what about vomiting, diarrhoea, use other supplementations such vitamins, minerals, …)
The study will be a candidate to be published in the journal as an original article but after a significant correction.
Author Response
REVIEWER 4
Dear Editorial team and Reviewer
Thank you for making quick progress on my manuscript. We appreciate your time and effort in providing valuable feedback on our manuscript. We have been able to incorporate changes to reflect most of the suggestions provided by the reviewers.
- Reviewer comment:
The statistics lack information about p-value as statistical significant (so how should be interpreted sentence: ”Wilcoxon paired test (p < 0.05 and p < 0.1).
Response:
The definition of statistically significant difference was added in the statistical analysis section. (Line 219-220)
- Reviewer comment:
Lack of information about the program used to evaluate the nutrition intake.
Response:
We add more information about this program. (Line 141-145)
- Reviewer comment:
Significant results, connected with decreased butyric acid level after 50 days of using a placebo (5.04 ± 2.64 vs 3.56 ± 2.32; p=0.023) – have been without complete interpretation in DISCUSSION. (“In contrast, in the placebo group, butyric acid was decreased. This indicates that there was no improvement in the gut environment of the placebo.” Such a result suggests that maybe using milk as a placebo is not suitable for the development of infants.
Response:
Thank you for pointing this out. The decreased butyric acid in the placebo group may be due to the high number of potentially pathogenic bacteria (Enterobacteriaceae) and the reduced intake of fibre, which acts as a substrate for producing SCFA by beneficial bacteria. This explanation was added to the discussion section (Line 461-464).
- Reviewer comment:
The paper lacks a paragraph entitled LIMITATIONS of study (at the end of DISCUSSION). In this paragraph, it should be included, for example – that the daily food intake of both groups was recorded using food records (what about vomiting, diarrhoea, use other supplementations such vitamins, minerals, …)
Response:
Thank you for your suggestion. During the intervention period, all subjects were also asked to fill out a schedule of product consumption, side effects after consuming the product (if any), stool quality (time of defecation, frequency of defecation, stool colour, and Bristol stools), and drug consumption (if any). There were no reports from the subjects regarding the side effects (vomiting or diarrhoea), even though some subjects had a common cold and were taking some drugs (paracetamol). It might be confounding factors that affect the results and will be considered as our research limitation. (Line 491-498)
Round 2
Reviewer 1 Report
The main concerns were corrected by the authors in this new version of the manuscript.
Reviewer 2 Report
According to the author guidelines (https://www.mdpi.com/journal/nutrients/instructions), “authors are required to pre-register clinical trials with an international clinical trials register and cite a reference to the registration in the Methods section. Suitable databases include clinicaltrials.gov, the EU Clinical Trials Register and those listed by the World Health Organisation International Clinical Trials Registry Platform.
Approval to conduct a study from an independent local, regional, or national review body is not equivalent to prospective clinical trial registration. MDPI reserves the right to decline any paper without trial registration for further peer-review.”
The statistical power was too low for this study to contribute to the accuracy of results and scientific soundness.
Reviewer 4 Report
I believe the manuscript has been sufficiently improved to warrant publication in Nutrients.